# Automated Detection of Premature Flow Transitions on Wind Turbine Blades Using Model-Based Algorithms

**Ann-Marie Parrey *** , **Daniel Gleichauf** , **Michael Sorg** and **Andreas Fischer**

Bremen Institute for Metrology, Automation and Quality Science, University of Bremen, 28359 Bremen, Germany; d.gleichauf@bimaq.de (D.G.); m.sorg@bimaq.de (M.S.); andreas.fischer@bimaq.de (A.F.)

\* Correspondence: am.parrey@bimaq.de

**Abstract:** Defects on rotor blade leading edges of wind turbines can lead to premature laminar–turbulent transitions, whereby the turbulent boundary layer flow forms turbulence wedges. The increased area of turbulent flow around the blade is of interest here, as it can have a negative effect on the energy production of the wind turbine. Infrared thermography is an established method to visualize the transition from laminar to turbulent flow, but the contrast-to-noise ratio (CNR) of the turbulence wedges is often too low to allow a reliable wedge detection with the existing image processing techniques. To facilitate a reliable detection, a model-based algorithm is presented that uses prior knowledge about the wedge-like shape of the premature flow transition. A verification of the algorithm with simulated thermograms and a validation with measured thermograms of a rotor blade from an operating wind turbine are performed. As a result, the proposed algorithm is able to detect turbulence wedges and to determine their area down to a CNR of 2. For turbulence wedges in a recorded thermogram on a wind turbine with CNR as low as 0.2, at least 80% of the area of the turbulence wedges is detected. Thus, the model-based algorithm is proven to be a powerful tool for the detection of turbulence wedges in thermograms of rotor blades of in-service wind turbines and for determining the resulting areas of the additional turbulent flow regions with a low measurement error.

**Keywords:** image processing; pattern recognition; wind energy turbines; turbulence wedges

## 1. Introduction

### 1.1. Motivation

Electrical energy created by wind turbines has become an increasingly important part in providing clean power. However, wind turbines are exposed to many environmental influences (e.g., hail [1], rain or insects [2]) that contribute to defects such as erosion and contamination of the rotor blade, especially at the leading edge. Defects influence the geometry and surface quality of the rotor blade and may lead to premature transitions from laminar to turbulent flow in the boundary layer. Premature transitions create distinctly wedge-shaped areas of turbulent flow in otherwise laminar flow regions, the so-called turbulence wedges [3]. The increase in the overall surface area with turbulent flow due to the existence of turbulence wedges can amplify acoustic emissions [4] as well as aerodynamic imbalances [5]. Furthermore, an increase in area with turbulent flow negatively affects the aerodynamic properties (i.e., decrease in lift, increase in drag), and thus reduces the annual energy production [6]. Therefore, it is necessary to monitor the condition of the blade's surface that influences the flow to quantify the amount of additional turbulent flow area. Infrared thermography is an established contactless, in-process measurement technique to visualize laminar and turbulent flow regions on operating wind turbines without blade modifications [7–9]. A temperature difference exists between different boundary layer flow regimes due to varying local heat transfer coefficients. Using this technique, areas with turbulent flow such as the turbulence wedges can be visualized. A heating of the rotor

surface is desirable to increase the temperature difference between the blade surface and fluid, which increases the contrast between laminar and turbulent flows. However, often, no active heating is available when operating wind turbines and the blade is only heated passively by sunlight. For this reason, an image processing algorithm is required which is capable of reliably detecting turbulence wedges under low contrast conditions. This would enable extensive field studies of the boundary layer flow state around the blade of operating turbines. In addition, the required algorithm has to determine the wedges' features such as position and size with a low uncertainty to finally quantify the resulting total area with turbulent flow.

### 1.2. State of the Art

Various methods exist for detecting laminar–turbulent flow transitions in thermograms, yet most approaches explicitly ignore turbulence wedges. One such approach called 'local infrared thermography' is presented by Mertens et al. [10] to detect unsteady transitions in periodic pitching processes. A pitching airfoil was investigated in a wind tunnel and thermograms were captured over multiple pitching periods. The airfoil was externally heated with a spotlight to increase the temperature difference between the airfoil and boundary layer flow. To detect the flow transition, the intensities of each pixel were detected over time and assigned to the corresponding phase of the pitch angle. Extrema in the intensity signal mean that the transition is passing through the current location. While the method successfully detects the flow transition, the analysis is not applicable to single thermograms, which is often the only type of data available from field measurements of in-service wind turbines. As mentioned before, the external active heating is also not often realized in field measurements.

Crawford et al. [11] located the flow transition in a crossflow-dominated environment on a heated swept wing model. The wing model was tested in wind tunnel and flight experiments, where the model was mounted on an airplane. The crossflow-dominated environment produces a jagged transition front, also called a sawtooth transition pattern. This sawtooth pattern is similar to turbulence wedges, albeit on a smaller scale. Thus, the measurement task is similar to detecting turbulence wedges. Furthermore, one transition front did in fact include one turbulence wedge. The heating of the model was realized with internal electrical heating wires. A special coating which reduced reflections, combined with the heating, improved the contrast-to-noise ratio (CNR) between laminar and turbulent flow regions in the thermograms. The saw-toothed transitions were detected through local maxima in the intensity gradient profiles after a series of image filters. Afterwards, a statistical analysis was performed to determine a prominent transition position of the sawtooth transition pattern. As the focus was put on estimating a transition position from the sawtooth pattern, turbulence wedges were intentionally rejected by the analysis. This analysis works with single thermograms but utilizes many filters and processing steps to reach its results, which have to be adjusted and optimized. Furthermore, the heating of the blade and the coating of the blade cannot be realized in field measurements of wind turbines in motion.

To detect the natural flow transition onset and end on helicopter rotor blades, Richter and Schülein [12] analyzed thermograms of rotating blades. They investigated a model blade on a whirl tower as well as full-scale rotor blades on helicopters during ground run and hovering. To increase the temperature differences between the blade surface and fluid, the blade was either spun fast to cool it, cooled with ice or passively heated in sunlight. In the thermograms, intensity profiles in the direction of the chord of the blade were examined, omitting all chord-wise positions of premature transitions. The intensity profiles consist of linear sections with different slopes. The three distinct slopes can be attributed to the laminar, transitional and turbulent regions of the boundary layer flow. Each slope was estimated with a linear function. The intersection point of the linear estimate of two adjacent regions was then classified as the transition onset and transition endpoint, respectively. The natural transition was successfully detected even for low signal-to-noise

ratios, which resulted from short exposure times. However, as turbulence wedges were excluded from the analysis, premature flow transitions were not detected.

Dollinger et al. [13] investigated thermograms of wind turbines in operation to determine the flow transition. To determine the transition position with subpixel accuracy, each chord-wise intensity profile in the thermographic image was approximated with the Gaussian error function. However, turbulence wedges were not detected reliably due to the low CNR between laminar and turbulent flow regions. Similarly, Gleichauf et al. [14] used intensity gradients to also detect flow transitions in thermograms of a wind turbine in operation. Since chord-wise intensity gradients are not sufficient for the reliable detection of turbulence wedges, the thermogram was at first rotated to such a degree that the subsequently evaluated intensity gradient was perpendicular to the premature transition lines. As a result, the rotation increased the sensitivity for the flow transition detection with regard to premature flow transitions. However, turbulence wedges with low CNR to the surrounding laminar flow still remain a challenge.

The current algorithms for detecting premature transitions only make use of image information such as intensities, which are often evaluated in single pixel lines without context or comparison to neighboring image parts. Therefore, when premature transitions are detected, they account for single points in the transition front. However, neighboring premature transition points are not grouped together. Not recognizing turbulence wedges as a whole complicates the quantification of the wedges' features, such as their sizes. Only Gleichauf et al. [14] and Dollinger et al. [13] quantify figure of merits related to additional turbulent area due to premature transitions, by calculating the differences between the positions of the natural and premature transition lines instead of adding up the turbulent area of turbulence wedges. Furthermore, pattern recognition has not been utilized for turbulence wedge detection by any of the discussed approaches. Pattern recognition would allow for the detection of each turbulence wedge as a whole, based on certain features such as their shape. An algorithm which specifically uses pattern recognition to reliably detect turbulence wedges even under low CNR conditions has not been reported yet, although such an algorithm seems promising for extracting and quantifying the wedges' features such as the additional turbulent area with low uncertainty.

*1.3. Aim and Outline*

An automated, model-based image processing algorithm is introduced, which reliably detects premature laminar–turbulent flow transitions in thermographic flow visualization images even in low-contrast scenarios. The fact that premature transitions lead to wedge-shaped areas of turbulent flow is incorporated in the algorithm with the use of wedge-shaped templates. These templates are then used to detect turbulence wedges and also to determine the positions and sizes of the turbulence wedges in thermograms. As a result, the additional amount of area with turbulent flow originating from premature transitions can be quantified. In order to characterize the capabilities of this approach, simulations as well as validation experiments on thermograms of in-process wind turbines are performed.

Section 2 contains the description of the novel image processing algorithm as well as a definition of the area with turbulent flow resulting from premature transitions. In Section 3, the implementation of the algorithm and the simulation and measurement setup are explained. The results of the verification of the algorithm for the simulation and the validation of real thermograms from in-process wind turbine measurements are presented and discussed in Section 4. Section 5 provides concluding remarks and an outlook.

## 2. Measurement Approach

*2.1. Thermogram Characteristics and Measurands*

The measurement quantities from a thermogram of a rotor blade with a premature flow transition are the position $x_i$ of the turbulence wedge and its size, consisting of the height $h_i$ and the width $w_i$, where $i$ is the running index of the wedge number. In Figure 1 (left), a real thermogram is shown with the marked position and size of one turbulence wedge as an

example. Note that the actual temperatures of the different flow regimes are irrelevant for the wedge detection algorithm, which is why all values in the thermograms are interpreted as intensities between 0 and 1 by normalizing with the maximum intensity $I_{max}$.

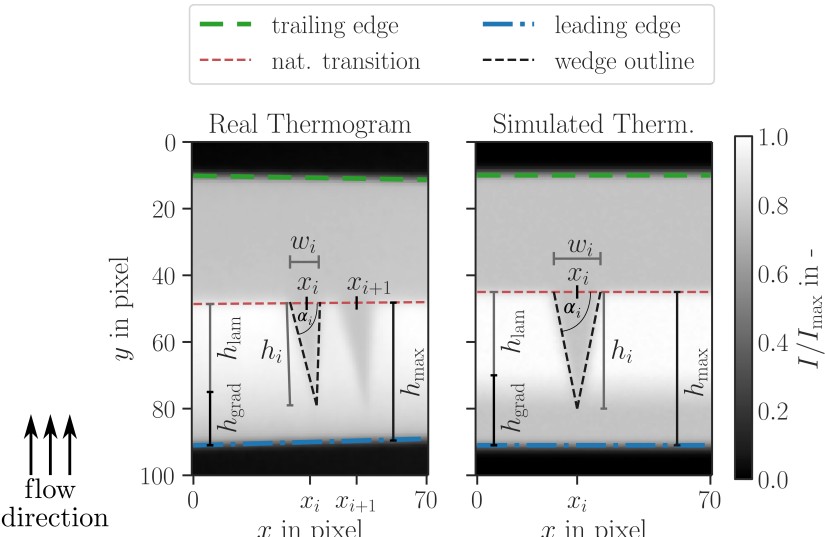

**Figure 1.** A real thermogram of a rotor blade of a wind turbine in operation with two turbulence wedges (**left**) and a simulated thermogram with one turbulence wedge (**right**). In this case, the rotor blade is warmer than the surrounding fluid, which means that the rotor blade is cooled by the boundary layer flow. The cooler the temperature, the lower the pixel intensity. Important characteristics of the thermograms are indicated.

According to Figure 1, the thermogram shows the leading edge and the trailing edge of the rotor blade, as well as the natural transition line that indicates the position of the non-premature flow transition from laminar to turbulent. The region of interest for the detection of turbulence wedges lies between the leading edge and the natural transition line, i.e., their distance equals the maximal height $h_{max}$ of a turbulence wedge. In general, $N \in \mathbb{N}$ turbulence wedges are visible in one thermographic image. The height $h_i$ of a turbulence wedge is defined as the orthogonal distance of the wedge's tip to the natural transition line, i.e., the wedge's base. The width $w_i$ is defined as the width of the base. The position of the turbulence wedge is defined as $(x_i, y_i), i \in \{1, \dots, N\}$, where $x_i$ is the center of the wedge's base. The $y$-position of the wedge's base is set to $y_i = y_{NT}(x_i)$, where $y_{NT}(x_i)$ denotes the $y$-position of the natural transition line at $x_i$. The position $x_i$ together with the height $h_i$ and the width $w_i$ of each turbulence wedge are the measurement quantities returned by the wedge detection algorithm. Furthermore, the height and the width are used to calculate the area $A_i$ of the turbulence wedge, with which the additional turbulent area due to premature transitions is quantified.

One further characteristic feature of the turbulence wedge is its symmetry or the lateral tip position. The tips of turbulence wedges in real thermograms are often not centered below the middle of their base. An example is shown in Figure 1 (left), where the wedges are skewed to the right. The wedges can appear skewed due to the camera perspective or due to the rotational movement of the rotor blade, which implies crossflow. The tip's position can be described using the angle $\alpha_i$ between the left edge of the turbulence wedge and the wedge's base. While the tip's position does not influence the size of the area of the turbulence wedge, it still needs to be taken into account for the detection of the turbulence wedge with the model-based algorithm.

A detailed analysis of real thermograms revealed an intensity gradient in the region between the leading edge and natural transition line, which is characterized by decreasing pixel intensities towards the leading edge. Due to the curvature of the blade at the leading edge and the dependency of the emission on the observation direction [15], lower pixel

intensities were captured in the thermogram at the leading edge. Furthermore, the heat transfer coefficient in the laminar flow regime was higher near the leading edge, where the laminar boundary layer was not yet fully developed, which also led to lower pixel intensities. The resulting intensity gradient in the thermogram begins at the leading edge and extends over a height $h_{\mathrm{grad}}$. The remaining flow region to the natural transition line, with the height $h_{\mathrm{lam}}$ has an almost constant intensity in the thermographic image. Since the intensity gradient near the leading edge reduces the image contrast of the turbulence wedge's tip, the accurate detection of the wedge's tip is challenging.

In order to verify and characterize the functionality of a wedge-detection algorithm, a ground truth needs to be known. Therefore, simulated thermograms are necessary, which emulate the characteristic features of real thermograms for given values of the measurement quantities. A simulated thermogram is shown in Figure 1 (right), which is not an exact replication of the real thermogram to its left but models all important characteristics. It shows only a single wedge with a different angle $\alpha_i$ compared to the real thermogram to illustrate the relation between the angle and the tip position of the wedge. Furthermore, the intensity gradient in the simulated thermogram is exaggerated compared to the real thermogram for the sake of visibility. A detailed description of the simulation of thermograms follows in Section 3. Simulated thermograms are already utilized in the subsequent description of the functionality of the algorithm principle for the wedge detection.

### 2.2. Determination of the Positions of the Blade Edges and the Natural Transition Line

Turbulence wedges are located in the region between the leading edge and the natural transition line. In order to focus the turbulence wedge detection on this region, the natural transition line and the leading edge need to be located in the thermogram first. To detect the $y$-positions of the blade edges and the natural transition line, a chord-wise gradient approach is chosen: for each image column, which corresponds to the chord-wise $y$-direction, the spatial derivative $\mathrm{d}I/\mathrm{d}y$ of the pixel intensities $I$ is calculated and then normalized with its maximum value:

$$\left(\frac{\mathrm{d}I}{\mathrm{d}y}\right)_{\mathrm{norm}} = \left|\frac{\mathrm{d}I}{\mathrm{d}y}\right| \cdot \left(\max\left(\left|\frac{\mathrm{d}I}{\mathrm{d}y}\right|\right)\right)^{-1}. \tag{1}$$

After the normalization, all local maxima above a threshold value of $0.1\,\mathrm{pixel}^{-1}$ are located. This threshold value is heuristically based as it suppresses noise but allows the local maxima due to the natural transition to be detected. A lower threshold value would lead to more false detections due to noise; a higher value would miss more actual transition points. To illustrate how the local maxima correspond to the $y$-positions of the blade edges and the natural transition, a simulated thermogram with a single turbulence wedge is shown in Figure 2a, for which the gradient is calculated for two different image columns. The studied image columns are chosen so that one (marked with a blue dotted line) intersects the turbulence wedge and therefore contains a premature transition. The other (marked with a dashed orange line) does not intersect a turbulence wedge and thus contains a natural flow transition.

According to Figure 2a, the pixel intensities of the background and the blade differ significantly, which leads to large local maxima in the gradient at the positions of the leading and the trailing edge, respectively. The natural transition also leads to a local maximum of the intensity gradient due to a large difference in pixel intensities between the laminar and the turbulent flow regions. Therefore, three local maxima can be present in the intensity gradient, corresponding to the trailing edge, the natural transition and the leading edge, see Figure 2b, at the top. The $y$-positions of the maxima are set as the $y$-positions of the blade edges and the natural transition line. In cases where only two maxima exist, see Figure 2b, at the bottom, the maxima are assigned to the trailing and leading edges. This case occurs when the chord-wise intensity profile contains a premature transition, which leads to intensity gradients that are below the threshold. Hence, the natural transition is not

detectable in image columns where a premature transition exists. Therefore, the detected points of the natural transition are fitted with a linear regression so that the $y$-position $y_{\text{NT}}$ of the natural transition is defined for each $x$-position of the thermogram and also for the region of the premature flow transition.

With the extracted $y$-positions of the natural transition line and the leading edge, the wedge detection algorithm can now be applied to the thermogram.

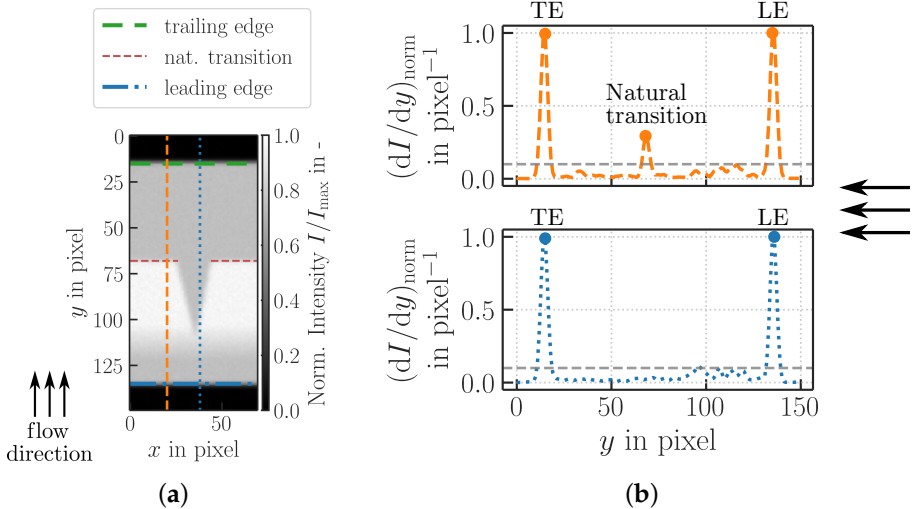

**Figure 2.** Simulated thermogram (**a**) and normalized gradient (**b**) for two chord-wise positions. The blue line intersects a turbulence wedge, which results in less pronounced maxima of the gradient below the threshold of $0.1\,\text{pixel}^{-1}$, marked with a dashed gray line. (**a**) Simulated thermogram with one turbulence wedge. The trailing edge is labeled with a green dashed line, the leading edge with a blue dash-dotted line. The natural transition is marked with a red dashed line, excluding premature transitions. The two vertical lines mark the positions for the intensity gradients, see (**b**). (**b**) Intensity gradients normalized by their respective maximums for the two lines at $x_{\text{orange}} = 20\,\text{pixel}$ (**top**) and $x_{\text{blue}} = 38\,\text{pixel}$ (**bottom**). Local maxima are marked with dots and correspond to the trailing edge (TE), the natural transition and the leading edge (LE).

### 2.3. Wedge Detection Algorithm

In order to detect turbulence wedges in thermograms, a model-based algorithm is proposed, which uses a technique called template matching. Template matching finds image parts that match the used template, which means that the proper choice of the template on the basis of a priori knowledge is crucial for the functionality of the algorithm. As a preface, the wedge-shaped templates are described in Section 2.3.1. Then, the two parts of the wedge detection algorithm are explained: the detection of the turbulence wedge's position in Section 2.3.2 and the determination of the wedge's size in Section 2.3.3.

### 2.3.1. Wedge Template

Template matching uses a template whose shape is similar to the desired feature that is to be found in the image. Therefore, the template has a triangular shape with the three parameters height $h'$, width $w'$ and angle $\alpha'$—see Figure 3. The angle $\alpha'$ is used to set the position of the template's tip $(x_{\text{tip}}, y_{\text{tip}})$.

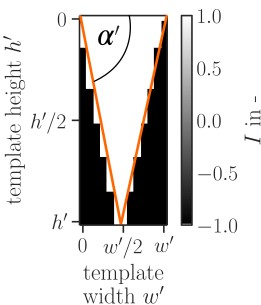

**Figure 3.** Example of a turbulence wedge template with height $h'$, width $w'$ and angle $\alpha'$, which determines the position of the tip.

In the coordinate system of the template, the tip's $x$-position can be calculated by using the equation

$$x_{\text{tip}} = h' \cdot \cos(90 - \alpha'), \tag{2}$$

where the $y$-position of the tip equals the wedge height, i.e.,

$$y_{\text{tip}} = h'. \tag{3}$$

Note that the width of the wedge at the base is not changed when the position of the tip is adjusted to tailor the template to the turbulence wedge in the thermogram. Currently, the adjustment of the angle $\alpha'$ is carried out manually, because $\alpha'$ can be kept constant for all turbulence wedges in the studied thermographic image.

Each template pixel value $v$ inside the wedge was set to $v_{\text{wedge}} = 1$, while the surrounding parts were set to $-1$—see Figure 3. The objective of doing so is to penalize all image pixels that do not fit the wedge shape. Note that the height and the width values of the template are positive integer values, because a subpixel interpolation has not been considered yet. The described template is subsequently utilized in the wedge detection algorithm to determine the positions of the turbulence wedges as well as the areas of the turbulence wedges.

2.3.2. Detection of the Wedge Position

In order to detect the positions of the turbulence wedges, a cross-correlation of the thermogram with the wedge template is used:

$$C[x] = \sum_p \sum_q \iota_{p,q} \cdot \tau_{p-y_{\text{NT}}(x),\, q-x}, \tag{4}$$

where $\tau$ denotes the template and $\iota$ denotes the thermogram section with the same size of the template. The variable $p$ is the row index of each image matrix, while $q$ is the column index. In accordance with the natural occurrence of turbulence wedges on a rotor blade, the template's $y$-position equals the $y$-position $y_{\text{NT}}(x)$ of the natural transition line, leading to a single cross-correlation result $C[x]$ for each $x$-position. The process of the detection of the wedges' positions is exemplified in Figure 4 using a simulated thermogram with three wedges, shown in Figure 4a.

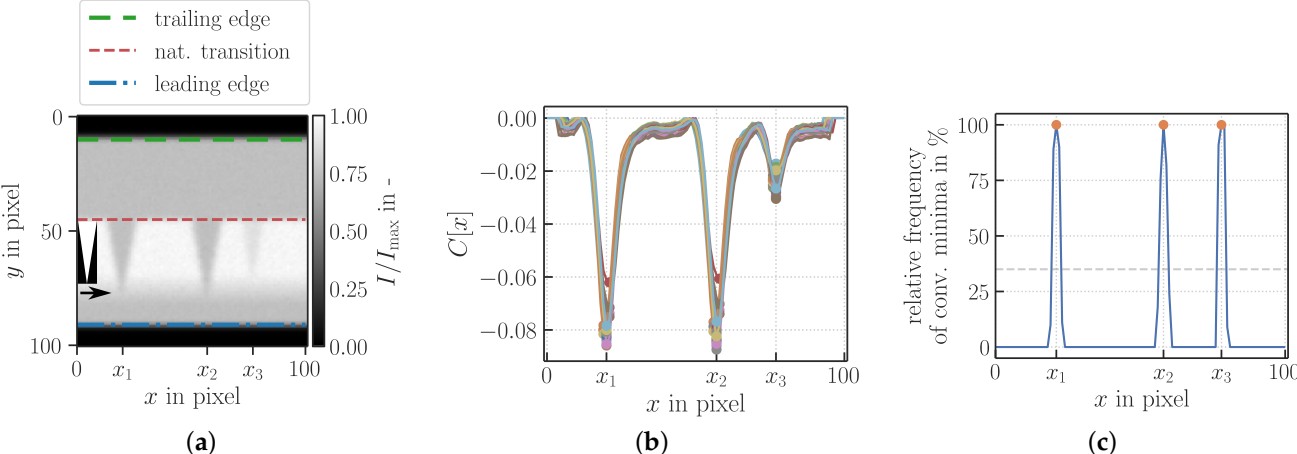

**Figure 4.** Illustrative analysis of a simulated thermogram (**a**) to detect the turbulence wedges' positions. The position detection is implemented using cross-correlations with randomized templates (**b**). Maxima in the relative frequency of minima in the cross-correlation results (**c**) determine the position of the turbulence wedges. (**a**) Simulated thermogram with three differently sized wedges. The wedge on the right has a lower contrast. The rectangle on the left side represents a template in its starting position, with an arrow below indicating the direction in which it is moved pixel-wise during the cross-correlation. (**b**) Cross-correlation $C[x]$, see Equation (4), of the simulated thermogram in (**a**) and $J = 100$ templates of varying sizes. Each color corresponds to a template of a different size. In this example, the smallest template has a size $(h', w')$ of $(27, 8)$ pixel; the largest $(41, 16)$ pixel. Local minima are marked with a dot. (**c**) Relative frequency of minima in the results of the cross-correlations. Each position $x_i$ of a maximum (orange dot) is marked. The cut-off percentage of 35% is marked with a dashed gray line.

A local minimum in the results of the cross-correlation indicates a good agreement between the thermogram and the template. To enable the detection of turbulence wedges with different, yet unknown sizes, the cross-correlation calculation is repeated for $J \in \mathbb{N}$ templates with different wedge heights and widths. Each template size corresponds to a different curve in the cross-correlation results shown in Figure 4b. For each $x$-position, the number of local minima (marked as dots) in the cross-correlation results is counted and normalized by the total number of studied templates, which results in a relative frequency of minima for each $x$-position, see Figure 4c. If the relative frequency is larger than 35% at a certain $x$-position, which is an empirically based value, it is considered to be a likely position for a wedge. The so determined $x$-positions $x_i, i \in \{1, \ldots, N\}$, correspond to the middle of the wedges' bases and thus to the wanted $x$-position of each turbulence wedge.

In addition, at each of the found position $x_i$, the minimum of all cross-correlation values is determined. The associated template $\tau_0$ with the height $h_{0,i}$ and a width $w_{0,i}$ has the best fit to the underlying wedge. The found wedge positions $x_i$ as well as the best fitting template $\tau_{0,i}$ for each $x_i$ position with $(h_{0,i}, w_{0,i})$ are needed for the precise determination of the wedges' areas, which is the next step of the algorithm.

### 2.3.3. Determination of the Wedge Area

In order to determine the additional area with turbulent flow in a thermogram caused by premature flow transitions, the area $A$ of all $N$ wedges in a thermogram needs to be known. The total area $A$ is determined by summing up the area $A_i$ of each wedge:

$$A = \sum_{i=1}^{N} A_i. \tag{5}$$

The area $A_i$ of each turbulence wedge in the thermogram is calculated with the formula

$$A_i = \frac{1}{2} h_i w_i, \tag{6}$$

where $h_i$ denotes the height and $w_i$ the width of the wedge at the position $x_i$. In order to determine $h_i$ and $w_i$, a measure of similarity between a template at the position $x_i$ and the thermogram section of the same size is required, which is calculated for different template sizes but at a fixed position to finally identify the most fitting template size.

The chosen measure of similarity is the weighted correlation, which is selected instead of the regular correlation due to the intensity gradient near the leading edge in the thermogram, which leads to a low contrast between the wedge's tip and the surrounding region. This low contrast between tip and surrounding causes the wedge height to be estimated too small and therefore leads to an erroneously determined wedge area. By using the weighted correlation, which weights a good fit between the template and the wedge at the base stronger than at the tip, the intensity gradient near the wedge's tip is counteracted. The weighted correlation in between a template $\tau$ and the thermogram section $\iota$ is defined as

$$\text{corr}(\tau, \iota) = \frac{\text{cov}(\tau, \iota)}{\sqrt{\text{cov}(\tau, \tau)\text{cov}(\iota, \iota)}}, \tag{7}$$

the covariance is the weighted covariance according to

$$\text{cov}(\tau, \iota) = \frac{\sum_p \sum_q \gamma_{pq} \cdot (\tau_{pq} - \overline{\tau})(\iota_{pq} - \overline{\iota})}{\sum_p \sum_q \gamma_{pq}}, \tag{8}$$

with the weight matrix $\gamma$ and the weighted mean values

$$\overline{\tau} = \frac{\sum_p \sum_q \gamma_{pq} \tau_{pq}}{\sum_p \sum_q \gamma_{pq}}, \quad \overline{\iota} = \frac{\sum_p \sum_q \gamma_{pq} \iota_{pq}}{\sum_p \sum_q \gamma_{pq}}. \tag{9}$$

The index $p$ indicates the row index of each matrix, while $q$ is the column index.

To determine the height $h_i$ and the width $w_i$ of the wedge at the position $x_i$, the template $\tau_{0,i}$ with the height $h' = h_{0,i}$ and the width $w' = w_{0,i}$, which was determined during the detection of the wedges' positions in Section 2.3.2, is used as an initial wedge size for the search. Since the sizes of the templates used in the position detection are chosen randomly, it is not ensured that $h_{0,i}$ and $w_{0,i}$ are the actual height and width of the turbulence wedge at the position $x_i$. Therefore, additional templates with sizes in the vicinity of the initial template size are investigated by calculating the weighted correlation according to Equations (7)–(9). The maximum value of these correlations yields the optimal size parameters $(h_i, w_i)$ of the wedge at the position $x_i$, which are then inserted into Equation (6) to obtain the wedge area.

## 3. Implementation of the Model-Based Algorithm and the Experimental Setup

### 3.1. Numerical Implementation of the Algorithm

The model-based wedge detection algorithm is implemented in the programming language *Python*. The steps to determine the position $x_i$, the size parameters $(h_i, w_i)$ and the area $A_i$ of each turbulence wedge are outlined in a flowchart in Figure 5.

As a first step, either the simulated thermogram is created or the recorded thermogram of the wind turbine blade is loaded. The loaded thermogram is then preprocessed, which includes an optional normalization of the thermogram intensities by the maximum intensity as well as an image rotation. The rotation aligns the natural transition line of the thermogram horizontally, which simplifies the calculation of the correlations as the $y$-positions of the line remain constant. To determine the rotation angle, the $y$-positions of the natural transition line $y_{\text{NT}}$ are at first detected in the unprocessed thermogram, using the classical gradient-based method described in Section 2.2. Using the coordinates of the natural transition line, the rotation angle $\varphi$ is calculated according to

$$\varphi = \arctan\left(\frac{y_{\text{NT}}(x_{\text{end}}) - y_{\text{NT}}(x_0)}{x_{\text{end}} - x_0}\right), \tag{10}$$

where $y^{NT}(x_{end})$ and $x_{end}$ are the last coordinates of the natural transition line and $y^{NT}(x_0)$ and $x_0$ are the first coordinates of the natural transition line in the original, i.e., non-rotated thermogram. The whole thermogram is rotated by the angle $\varphi$ using the *rotate* functionality from the *Python Image Library* (PIL). Note that the image rotation is not necessary for simulated images, where the natural transition line is already aligned horizontally.

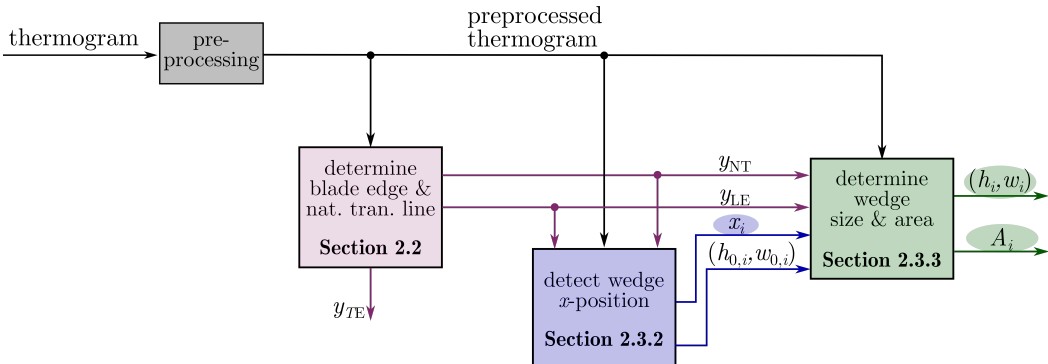

**Figure 5.** Flowchart of the model-based algorithm. $y_{TE}$ are the $y$-positions of the trailing edge, $y_{NT}$ the $y$-positions of the natural transition line and $y_{LE}$ the leading edge's $y$-positions. The output quantities, which are highlighted with a colored circular background, are the $x$-position $x_i$, the height $h_i$, the width $w_i$ and the area $A_i$ of each turbulence wedge, where $i$ denotes a running wedge number.

The preprocessing also includes the determination of the angle $\alpha_i$ of the turbulence wedges, which is currently the only manually estimated input parameter for the wedge detection algorithm. However, $\alpha_i$ remains the same for all turbulence wedges until the flow condition changes at which the thermograms are captured.

After the preprocessing, the $y$-positions of the leading edge $y_{LE}$ and the trailing edge $y_{TE}$, as well as the natural transition line $y_{NT}$, are detected in the thermogram by using the gradient-based method introduced in Section 2.2. The values obtained by the gradient-based method are fitted with a linear regression using the *RANSAC* algorithm of *Python's sklearn* library. To use the leading edge and the natural transition line in the wedge detection algorithm, the $y$-position values are rounded to the next integer. With the $y$-positions of the leading edge and the natural transition line, the maximum height $h_{max}$ of the turbulence wedges is calculated according to

$$h_{max} = \langle y_{LE}(x) - y_{NT}(x) \rangle, \tag{11}$$

where the mean is taken over all $x$-positions.

As the next step, the wedges and their $x$-positions $x_i$ are detected along the natural transition line by cross-correlating the thermogram with the wedge-shaped templates, see Section 2.3.2. For the position detection, $J \in \mathbb{N}$ differently sized templates with running template index $j$ are randomly generated, where $J = 100$ is the default value. The templates' heights $h'_j$ are drawn from a uniform distribution in the interval $[0.5 \cdot h_{max}, 0.95 \cdot h_{max}]$. This interval has proven adequate for detecting wedges in real thermograms, but the interval can also be adapted to any values in $(0, h_{max}]$ if needed. The investigation of real thermograms (see Section 3.2) has further shown that the height-to-width ratio of real turbulence wedges is, on average, $\langle h_i/w_i \rangle = 3$. Consequently, the templates' widths are drawn from a normal distribution with $\mathcal{N}(\mu_{w'} = h'_j/3, \sigma^2_{w'} = 0.2)$, where $\mu_{w'}$ is the mean and $\sigma^2_{w'}$ is the variance of the normal distribution.

To calculate the cross-correlation between the thermogram and each turbulence wedge template, the template with index $j$ is placed with its top left corner at $x_0 = 0$ and $y = y_{NT}(x_0)$ as a starting point. A section of the full thermogram the same size as the template is required for the cross-correlation calculation. The columns (i.e., the width) of the thermogram section are chosen from the position $x$ to $(x + w'_j)$ of the full thermogram. The rows (i.e., the height) of the thermogram section are determined from the position

$y_{NT}(x)$ to $(y_{NT}(x) + h'_j)$. The cross-correlation of the thermogram section and the template is calculated using Equation (4) and the template is moved to the right by 1 pixel. This way, the cross-correlation value $C[x]$ is calculated for each $x$-position. However, when the right edge of the template reaches the right edge of the thermogram, the top left corner is at $x = w_{therm} - w'_j$, where $w_{therm}$ is the width of the thermogram. For the remaining $x$-positions the cross-correlation can not be calculated as the template would otherwise protrude outside the borders of the thermogram. Therefore, due to the different widths $w'_j$ of each template, the length of the cross-correlation curve of each template is different. To align the results of the cross-correlations of different templates, the cross-correlation values for each template $j$ are shifted to the right by $w'_j/2$ and the maximum cross-correlation value is subtracted. Using the aligned cross-correlation curves, the relative frequency of minima is determined. The positions of the local maxima above 35% in the relative frequency then correspond to the wedge positions $x_i$. The threshold value of 35% can be adapted depending on the desired sensitivity of the algorithm. The higher the threshold value, the more contrast the turbulence wedges need to have in order to be detected, which can lead to overlooked turbulence wedges. A lower threshold value, on the other hand, can result in an erroneous detection of noisy structures in the image as a turbulence wedge. Note that the relative frequency is currently not interpolated, i.e., the wedge positions $x_i$ are integer values.

As the last step of the wedge detection algorithm, the wedges' sizes $(h_i, w_i)$ and their areas $A_i$ are determined for each wedge position $x_i$ using a weighted correlation, see Section 2.3.3. For each $x_i$, the initial template $\tau_{0,i}$ with the size $(h_{0,i}, w_{0,i})$ is used to create a range of new templates. Each template is then compared to the wedge to find the template with the best fit to the turbulence wedge. The sizes of the new templates are taken from a range of

$$h_{0,i} - 3 \, \text{pixels} \leq h'_k \leq h_{0,i} + 3 \, \text{pixels} \tag{12}$$

for the height and

$$w_{0,i} - 3 \, \text{pixels} \leq w'_k \leq w_{0,i} + 3 \, \text{pixels} \tag{13}$$

for the width. For each height and width combination of the two intervals, a new template is created, which results in $K = 49$ templates in total with running template index $k$. Using each template, the weighted correlation with the turbulence wedge at the position $x_i$ is calculated. To calculate the weighted correlation, the top left corner of the current template matrix is placed at $(x_i - w'_k/2)$ along the natural transition line at $y_i = y_{NT}(x_i)$, resulting in the top right corner of the template to be positioned at $(x_i + w'_k/2)$. The columns (i.e., the width) of the thermogram section are selected from the $x$-position $(x_i - w'_k/2)$ of the thermogram to $(x_i + w'_k/2)$. The rows (i.e., the height) of the thermogram section are determined from the $y$-position $y_{NT}(x_i)$ to $y_{NT}(x_i) + h'_k$. Thus, the thermogram section and the template have the same size. The weighted correlation is chosen as a measure of similarity so that the intensity gradient near the leading edge of the thermogram is counteracted, which is implemented through the weights $\gamma$ in the weight matrix. Thus, the weighting emphasizes a good match at the base of the template to the turbulence wedge. Therefore, the weights are larger near the base of the turbulence wedge, with $\gamma_{base} = 10$, and linearly decrease row by row to smaller weights near the tip, $\gamma_{tip} = 1$. The number of steps for the linear decrease, which was implemented with the *linspace* function of *Python's numpy*, is equal to the height $h'_j$ of the current template. The linear decrease is repeated in each matrix column for $w'_j$ columns. The weight matrix, the thermogram section and the template therefore all have the same size, and the weighted correlation is calculated using Equation (7). The calculations result in a phase space of correlation values, i.e., one value for each height-width combination, see Figure 6. The size of the template which attains the

highest correlation value, see red x in Figure 6, is the size $(h_i, w_i)$ of the turbulence wedge at position $x_i$, from which the area $A_i$ can be calculated with Equation (6).

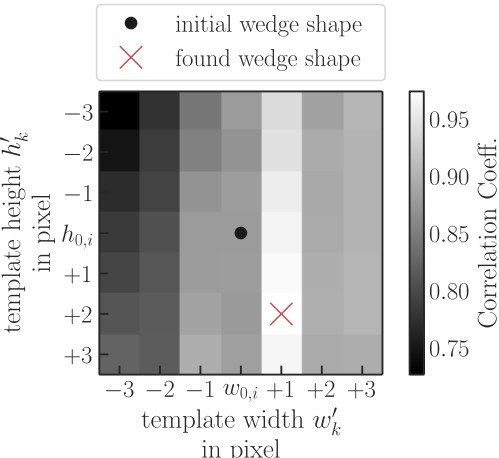

**Figure 6.** Phase space of correlations for a wedge at position $x_i$, over the template height $h'_k$ and width $w'_k$. The height $h_{0,i}$ and the width $w_{0,i}$ are the initial values for the determination of the wedge's size. The phase space point $(h_{0,i}, w_{0,i})$ is marked with a black dot. The point $(h'_k, w'_k)$ with the highest correlation value is marked with a red x.

### 3.2. Simulation Setup

In order to verify the wedge detection algorithm and characterize it with respect to the measurement uncertainties, simulated thermograms are needed, where all measurands of the thermogram are a priori known exactly.

The simulated thermogram consists of a matrix with a height $h_{\text{therm}}$ and a width $w_{\text{therm}}$. The trailing edge is placed at $y_{\text{TE}} = 0.10 \cdot h_{\text{therm}}$, the natural transition at $y_{\text{NT}} = 0.45 \cdot h_{\text{therm}}$ and the leading edge at $y_{\text{LE}} = 0.90 \cdot h_{\text{therm}}$. The blade edges and natural transition lines are aligned horizontally. The pixel intensities of the turbulent region $I_{\text{turb}}$ and the laminar region $I_{\text{lam}}$ are quantities that need to be modelled to values in real thermograms. The pixel intensity of the background is set to 0.1, as a low value results in a strong gradient to the blade edge. However, the pixel intensity of the background does not influence the wedge detection algorithm and the exact value is therefore inconsequential.

Moreover, the intensity gradient near the leading edge is simulated. Near the natural transition line, the laminar value is prevalent. Towards the leading edge, the pixel intensities start to linearly decrease at some $y$-position $y_{\text{LE}} - h_{\text{grad}}$, see Figure 1 (right). Then, at a $y$-position $y_{\text{LE}} - h_{\text{grad},0}$ near the leading edge, the decreasing intensity reaches the turbulent intensity value $I_{\text{turb}}$. After this $y$-position up to the leading edge, the pixel intensities are set to $I_{\text{turb}}$. The parameters $h_{\text{grad}}$ and $h_{\text{grad},0}$ are modelled according to real thermograms. Note that both parameters are subsequently presented normalized by $h_{\text{max}}$ to make them transferable to thermograms of any size.

In addition, the amount of Gaussian blur in the thermogram can be set, with a standard deviation $\sigma_{\text{blur}}$ of the Gaussian kernel of 1.0 pixel. The blur value was estimated manually to match the blur in real thermograms.

Furthermore, the noise in real thermograms is simulated by creating a matrix of noise values with the same size as the thermogram, for which each pixel value is drawn from a normal distribution $\mathcal{N}(\mu_{\text{noise}}, \sigma^2_{\text{noise}})$ with mean $\mu_{\text{noise}} = 0$ and variance $\sigma^2_{\text{noise}}$ in arbitrary pixel intensity units. By changing the variance $\sigma^2_{\text{noise}}$, the amount of noise in the image can be varied. The matrix is then added to the thermogram to obtain a noisy thermogram.

To emulate a turbulence wedge realistically, the parameters additionally required from real thermograms are the typical range of the height $h_i$ and the width $w_i$ of the turbulence wedge. Instead of the absolute height value (in pixels), the relative height $h_i / h_{\text{max}}$ is considered here, which makes the value applicable to thermograms of any size. To emulate the

average width of the turbulence wedges, the height-to-width ratio $h_i/w_i$ is further considered. The angle $\alpha_i$ of each wedges is set, by default, so that the tip position is centered below the middle of the wedge for all turbulence wedges. Finally, the typical contrast-to-noise ratio (CNR) of a turbulence wedge is emulated, where the CNR is defined as

$$\text{CNR}_i = \sqrt{\frac{(\langle I_{\text{lam}}\rangle_i - \langle I_{\text{wedge}}\rangle_i)^2}{\sigma(I_{\text{lam}})_i^2 + \sigma(I_{\text{wedge}})_i^2}},\tag{14}$$

where $\langle I_{\text{wedge}}\rangle_i$ and $\langle I_{\text{lam}}\rangle_i$ are the mean pixel intensities in the turbulent region inside the wedge and the laminar region right outside of the turbulence wedge with index $i$, respectively. Furthermore, $\sigma(I_{\text{wedge}})_i$ and $\sigma(I_{\text{lam}})_i$ denotes the spatial standard deviation of the pixel intensities of the turbulence wedge or its surrounding area.

To realistically emulate real thermograms, the above mentioned features of real thermograms and turbulence wedges need to be quantified. For this reason, real thermograms of wind turbines in operation are analyzed. In the analysis, 14 different thermograms and 43 different turbulence wedges are included. All heights and widths are measured using the program ImageJ [16], while the CNR calculations are performed in Python. The loaded thermograms are normalized to a range of $[0,1]$. The results can be seen in Table 1, where the first section contains the values that are important for the modelling of a thermogram without turbulence wedges. The second section contains the characteristic values of the turbulence wedges.

**Table 1.** Important characteristics of the thermograms of rotor blades (first section) and the turbulent wedges (second section) determined in real thermograms of wind turbines in motion. All thermograms were normalized to a range of 0 to 1 in arbitrary intensity units. The standard deviation of the mean is given as the measurement uncertainty.

| Characteristic | Determined Value | Simulated Value |
|---|---|---|
| $\langle I_{\text{lam}}\rangle$ | $0.96 \pm 0.01$ | 0.96 |
| $\langle I_{\text{turb}}\rangle$ | $0.75 \pm 0.02$ | 0.75 |
| $\langle \sigma(I_{\text{lam}})\rangle$ | $0.010 \pm 0.001$ | 0.009 |
| $\langle \sigma(I_{\text{turb}})\rangle$ | $0.009 \pm 0.001$ | 0.009 |
| $\langle h_{\text{grad}}/h_{\text{max}}\rangle$ | $0.50 \pm 0.07$ | 0.50 |
| $\langle h_{\text{grad},0}/h_{\text{max}}\rangle$ | $0.16 \pm 0.04$ | 0.15 |
| $\langle h_i/h_{\text{max}}\rangle$ | $0.73 \pm 0.02$ | $[0.6, 0.85]$ |
| $\langle h_i/w_i\rangle$ | $3.01 \pm 0.07$ | 3 |
| $\langle \text{CNR}_i\rangle$ | $10.1 \pm 1.1$ | $[2, 20]$ |

By adapting the mean values in Table 1 in the simulated thermograms, thermograms of rotor blades of in-service wind turbines are emulated realistically. The values used in the simulation can be seen in the rightmost column. Note that the values for $\langle \sigma(I_{\text{turb}})\rangle$ and $\langle \sigma(I_{\text{turb}})\rangle$ are identical in the simulation, because the marginal difference in the experimental results is neglected. Furthermore, the values of the heights $h_i/h_{\text{max}}$ of the simulated turbulence wedges are drawn from a uniform distribution in the interval $[0.6, 0.85]$ if not explicitly stated otherwise, which places the found mean value approximately in the middle of the range. The widths $w_i$ of the simulated turbulence wedges are drawn from a normal distribution $\mathcal{N}(\mu_w = h_i/3, \sigma_w^2 = 0.2)$, where $\mu_w$ is the mean and $\sigma_w^2$ is the variance of the normal distribution. By modifying the size of the turbulence wedges in the simulated thermograms in a small interval which is easily detectable by the algorithm, no size that is particularly well or poorly detected is chosen by chance.

In order to characterize the dependency of the model-based wedge detection algorithm on the CNR value as well as the size of the turbulence wedges, Monte Carlo simulations are performed which utilize the simulated thermograms. With the results of the Monte Carlo

simulations, the systematic and the random errors of the wedge position $x_i$, the wedge size $(h_i, w_i)$ and the wedge area $A_i$ are investigated.

For the investigation of the dependency of the algorithm on the CNR, the CNR of a single turbulence wedge is changed in a range of 2 to 20 by changing the pixel intensity of the turbulence wedge. The choice of the CNR range places the mean value of $\langle \text{CNR} \rangle = 10$ found in real thermograms in the middle of the range, but also covers more extreme CNR values. Varying the intensity of the turbulence wedges instead of changing the amount of noise in the image is more realistic, as the noise in real thermograms stays approximately constant throughout the image.

To investigate the dependency of the algorithm on the size of the turbulence wedges, the height of the turbulence wedge in the thermogram is changed systematically in a range of $0.1 \leq h_i / h_{\max} \leq 1.0$. Due to the identified relationship $\langle h_i / w_i \rangle = 3$, the wedge width is changed accordingly. The CNR value of the turbulence wedge, however, remains constant at $\text{CNR} = 11$ throughout this analysis.

### 3.3. Measurement Setup

Real thermograms of rotor blades of in-service wind turbines are required to validate the algorithm. Therefore, thermographic measurements are performed on a 1.5 MW wind turbine of the type GE 1.5 sl, manufactured by General Electric (Boston, MA, USA), with a hub height of 62 m and rotor diameter of 77 m. The thermograms are taken with an actively cooled infrared camera called imageIR 8300, manufactured by InfraTec GmbH (Dresden, Germany). This thermographic camera has an InSb focal plane array with a format of $(640 \times 512)$ pixel$^2$ where 1 pixel $\cong 15$ µm, and is sensitive to light of a wavelength of 2–5 µm. The dynamic range is 14 bit, the integration time is set to 1600 µs and the noise equivalent temperature difference is about 25 mK at 30 °C. The measurements are taken over multiple days on a wind turbines at a measurement distance of 100 m. Due to the length of the rotor blades and to improve the spatial resolution, a 200 mm telephoto lens is used. Consequently, the rotor blades are captured in segments. For this purpose, the thermographic camera is triggered externally with an optical trigger camera when the rotor blade is positioned horizontally, i.e., parallel to the ground. An example of a typical measurement setup can be seen in Figure 7, where thermograms of the suction side of the rotor blade are acquired.

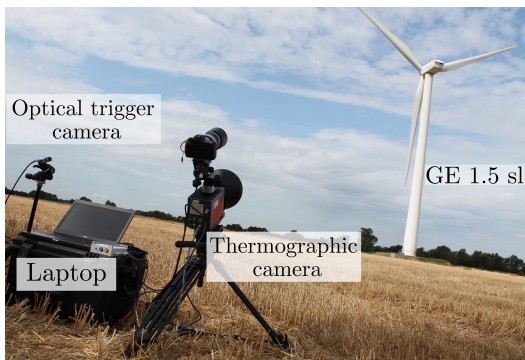

**Figure 7.** Photo of experimental setup for field measurements of wind turbines in motion. The thermographic camera, which takes thermograms of the wind turbine in the background, the optical trigger camera, which triggers the thermographic camera, and the laptop, which acquires the images, can be seen.

With the measurement setup, thermographic measurements of different rotor blade segments are taken, where the turbulence wedge have varying sizes as well as different CNR values, which are used to validate the wedge detection algorithm. Furthermore, the distance between wedges differs between the measurements, which has not been considered in the verification and which demonstrates the applicability of the wedge detection algorithm to real thermograms.

## 4. Results

In this section, the results of the image processing algorithm for turbulence wedge detection are presented. First, a verification and detailed characterization of the algorithm on simulated thermograms is performed in Section 4.1 and Section 4.2, respectively. Particularly, the dependency of the algorithm on the CNR value and the size of the turbulence wedge is investigated. The validation of the algorithm for measured thermograms of wind turbine blades in operation follows in Section 4.3. Note that the results of the verification and the validation with the model-based algorithm (MBA) are compared with the state-of-the-art, gradient-based image processing algorithm (GBA) from [14].

### 4.1. Verification

A verification of the model-based wedge detection algorithm is performed with a simulated thermogram with known position and size of the wedges. The studied thermogram contains three wedges, I, II and III, with the CNR values 19, 5 and 3. The simulated thermogram with the consecutively numbered turbulence wedges of different sizes is shown in Figure 8a, while the reference turbulence wedge outline as well as the CNR value are shown in Figure 8b. Both of the wedge detection algorithms result in a flow transition line which includes all premature transitions across the width of the thermogram, see Figure 8d. To determine the area of the turbulence wedges with the GBA, the $y$-coordinates of the natural transition line are subtracted from the $y$-coordinates of the detected premature transition line. The results of the GBA and the novel MBA are shown in Figure 8c,d, respectively. The GBA detects wedge I, but fails to detect the other two wedges with lower CNR values. In contrast, the MBA correctly detects all three wedges. The results of the verification thus show that, with the introduced model-based algorithm, a reliable detection even at CNR = 3 is possible, which is a significant improvement compared to the state-of-the-art, gradient-based algorithm.

A quantitative comparison of the areas found by the two wedge detection algorithms is shown in Table 2. The total area determined by the MBA has a relative deviation of 1.5% compared to the true value, while the state-of-the-art GBA detects a total area with a relative deviation of $-38.6\%$ from the true value due to the undetected turbulence wedges. The MBA determines a larger-than-true area for the smallest turbulence wedge in the middle, but smaller areas for the larger turbulence wedges, which seems to be worth investigating further in the subsequent, more detailed characterization.

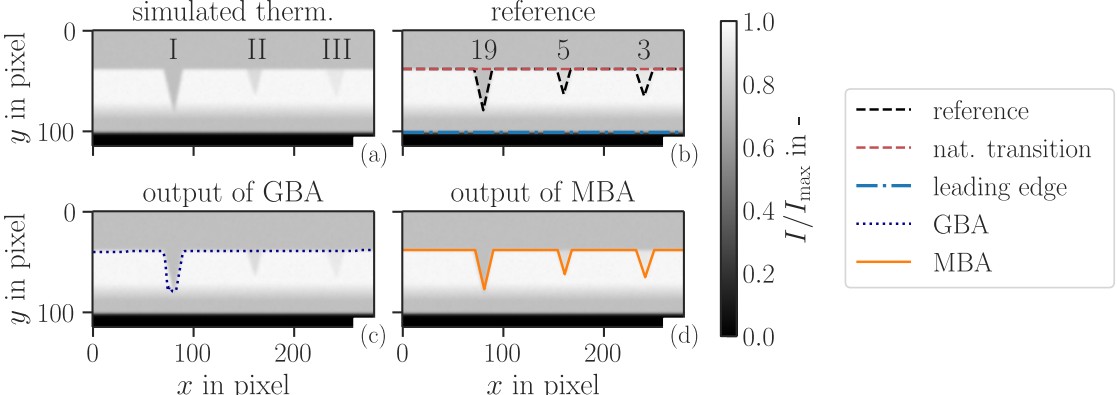

**Figure 8.** Simulated thermogram with three different turbulence wedges with differing intensities, leading to different CNR values. Both the model-based algorithm as well as the state-of-the-art, gradient-based algorithm were applied to the image. (**a**) shows the simulated thermogram with the turbulence wedges numbered with roman numerals. (**b**) shows the reference transition line as well as the CNR value of the wedges noted above the respective wedge. (**c**) shows the result of applying the gradient-based algorithm (GBA) of Gleichauf et al. [14] to the image and (**d**) shows the result of the application of the model-based algorithm (MBA) on the simulated thermogram.

**Table 2.** Areas determined by the model-based algorithm (MBA) as well as the state-of-the-art, gradient-based algorithm (GBA) for the wedges in Figure 8. The area values are given in unit pixel$^2$.

|  | True Value | GBA | MBA |
|---|---|---|---|
| Wedge I | 389.5 | 512.0 | 370.5 |
| Wedge II | 187.5 | - | 180.0 |
| Wedge III | 256.5 | - | 270.0 |
| Total | 833.5 | 512.0 | 846.5 |
| Measurement error normalized by the true value in % | 0 | −38.6 | 1.5 |

### 4.2. Characterization

To characterize the model-based algorithm in more detail, Monte Carlo simulations with thermograms of the size $(140 \times 140)$ pixels$^2$ are utilized, with which the dependency of the model-based algorithm on the CNR value as well as on the size of the turbulence wedges is examined. The deviations of the detected number $N$ of turbulence wedges, the measured position $x_i$ and the size parameters $(h_i, w_i)$ as well as the area $A_i$ are examined.

First, the dependency of the measurement results on the CNR value is investigated. The CNR value is varied in a range of $2 \leq \text{CNR} \leq 20$ by changing the pixel intensity of the $\tilde{N} = 1$ turbulence wedge present in the simulated thermogram. For each CNR value, $n = 100$ thermograms are investigated. Note that the other parameters used in the simulation, such as the wedge size and the intensity gradient, are stated in Section 3.2.

The number $N$ of turbulence wedges that are detected in a thermogram is assessed and compared to the true value $\tilde{N}$. Here, the relative error

$$F_N = \frac{N_i - \tilde{N}_i}{\tilde{N}_i} \tag{15}$$

is investigated, where $N_i$ is the number of detected wedges in the $i$th thermogram and $\tilde{N}_i = 1$ is the true number of wedges. In Figure 9, the systematic error $\langle F_N \rangle$ is shown as a function of the CNR. Only for the lowest CNR value (CNR = 2), the number of detected wedges is not equal to the true number of wedges, where 7% of turbulence wedges are missed.

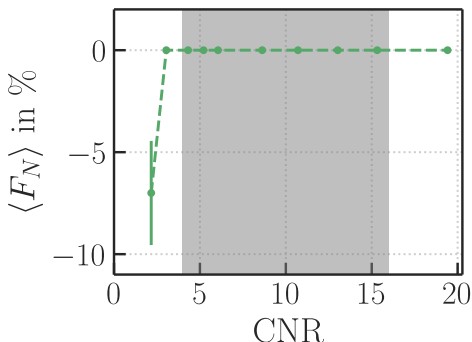

**Figure 9.** Mean relative error $\langle F_N \rangle$ for different CNR values. For each CNR value, $n = 100$ thermograms were investigated. The gray region marks CNR values found in real thermograms, CNR = $10 \pm 1\sigma$, with $\sigma = 6$. The error bar indicates the standard deviation of the mean.

In all other cases, $\langle F_N \rangle = 0$, which means that the number of found wedges is equivalent to the true number of wedges in the thermogram, i.e., the detection error rate is zero. Furthermore, the position error $\Delta x$ normalized by the average true width $\langle \tilde{w} \rangle$ of the wedges, is investigated:

$$\frac{\Delta x}{\langle \tilde{w} \rangle} = \frac{1}{\langle \tilde{w} \rangle}(x_i - \tilde{x}_i). \tag{16}$$

Here, $x_i$ is the $x$-position of the found wedge in the $i$th thermogram, $\tilde{x}_i$ is the true $x$-position and $\langle \tilde{w} \rangle = 15.01$ pixels is the average true width of all wedges used. Only the

$x$-position needs to be considered as the algorithm detects the wedges along the natural transition line, i.e., the $y$-position follows from the $x$-position and the determined natural transition line. Figure 10 shows $\frac{\Delta x}{\langle \bar{w} \rangle}$ as a function of the CNR. For the lowest CNR value of CNR $=2$, seven thermograms were excluded from the analysis as no turbulence wedge was found and, therefore, $x_i$ is unknown. The mean relative error $\langle \frac{\Delta x}{\bar{w}} \rangle$ stays between 2% and 4% for all CNR. The standard deviation of $\frac{\Delta x}{\bar{w}}$ in Figure 10 (right), which represents the random error, is larger than the systematic error for most CNR values by a factor of 1.5. However, for CNR $< 3$, the systematic error is larger than the random error. Therefore, the systematic error plays a significant role at low CNR $< 3$, where it should be corrected.

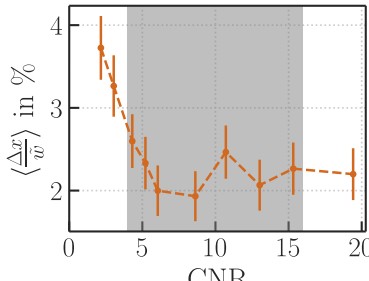 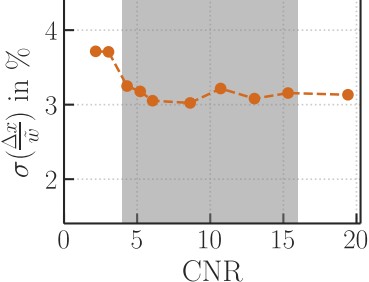

**Figure 10.** (**Left**): Mean position error $\left\langle \frac{\Delta x}{\bar{w}} \right\rangle$ normalized by the average wedge width (see Equation (16)) for $n = 100$ thermograms for different CNR values. The error bar indicates the standard deviation of the mean. The gray region marks realistic CNR values commonly found in real thermograms. (**Right**): The standard deviation of $\frac{\Delta x}{\langle \bar{w} \rangle}$ in percent, which represents the random error.

To investigate the systematic and random error concerning the size of the turbulence wedge, the relative deviation of the height, the width and the area of the wedge are assessed. The relative deviation for a variable $X$ is defined as

$$F_X = \frac{X_i - \tilde{X}_i}{\tilde{X}_i}, \tag{17}$$

where $X$ can be the height $h$, the width $w$ or the area $A$ to define the average relative deviations $F_h$, $F_w$ and $F_A$. Figure 11 shows the mean relative deviations of the heights (Figure 11a, left), the widths (Figures 11b, left) and the areas (Figure 11c, left) over different CNR values. In general, the lower the CNR, the larger the deviations. Furthermore, while the height of the wedge is consistently underestimated, the width is overestimated for all CNR values. The largest deviations occur at CNR $< 4$, which is a CNR value not often found in real thermograms as it is outside the $1\sigma$ interval. As a result, the turbulence wedge areas are underestimated throughout the investigated CNR range. The plots on the right hand side of each figure show the standard deviation of the relative deviation, which is a measure of the random error. For the width, the random error is slightly larger than the systematic error by a factor of about 1.5 for CNR $> 4$. However, for the height and the area, the random error is of about the same magnitude as the systematic error. Therefore, the results should be corrected by the systematic error in the future. Currently, the height and area of the wedge are underestimated on average by maximally about 10% for CNR values that are common in thermograms, $4 \leq$ CNR $\leq 16$.

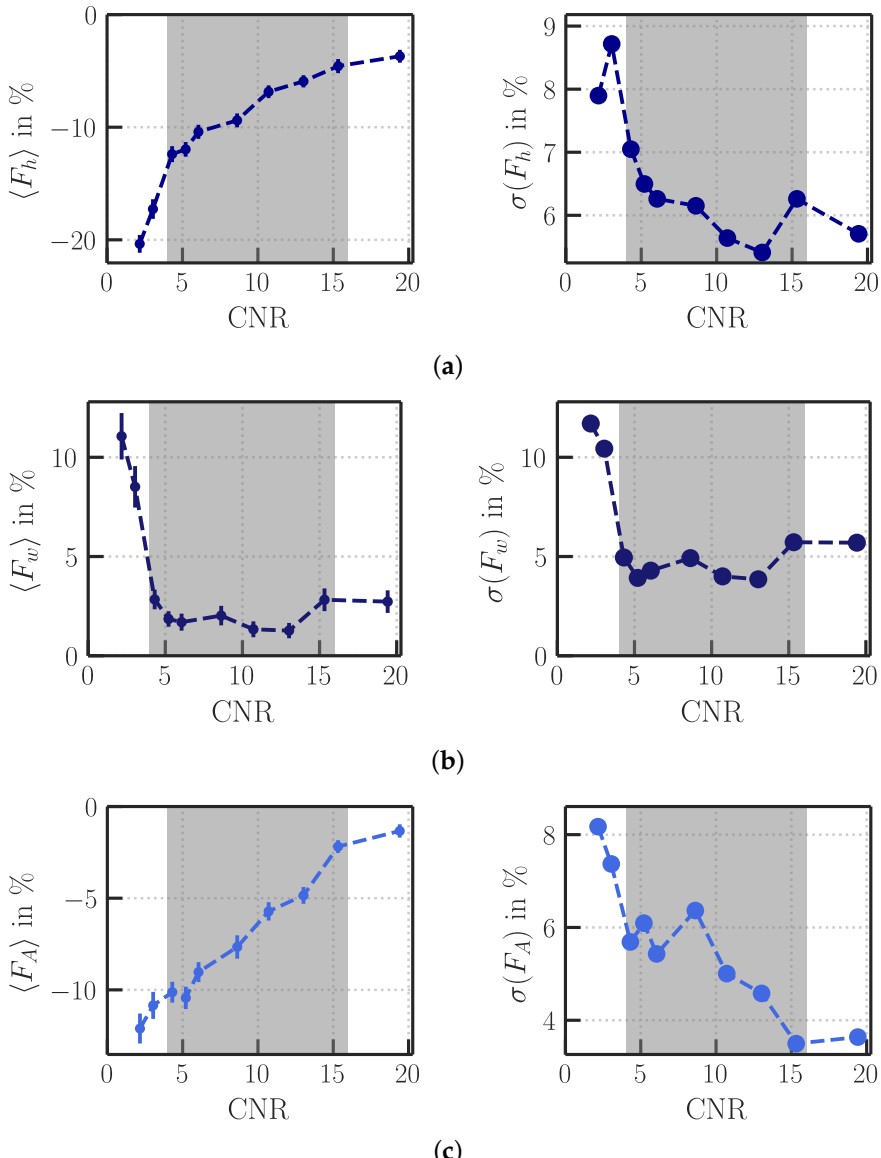

**Figure 11. Left**: Mean relative deviation of (**a**) the wedges' heights $\langle F_h \rangle$, the wedges' widths $\langle F_w \rangle$ (**b**) and wedges' areas $\langle F_A \rangle$ (**c**) for each CNR value with $n = 100$ trials with randomly sized wedges. The error bar indicates the standard deviation of the mean, and the gray region shows realistic CNR values found in real thermograms. **Right**: Standard deviation of all respective mean relative deviations in percent.

To further investigate how the height of the turbulence wedge influences the wedge detection algorithm, a thermogram of the size $(140 \times 140)$ pixels$^2$ with a single turbulence wedge with CNR $= 11$ is investigated, and the height $\tilde{h}$ of the wedge was changed systematically from $\tilde{h} = 0.1 \cdot h_{\max}$ to $\tilde{h} = h_{\max}$ in 15 steps. For each wedge height, $n = 50$ thermograms were investigated. The width of the wedge was drawn anew for each thermogram from $\mathcal{N}(\mu = \tilde{h}/3, \sigma^2 = 0.2)$. In Figure 12a, $\langle F_N \rangle$ is shown, while Figure 12b shows the position error, and Figure 12c shows the relative deviations of the wedge's size and area $\langle F_h \rangle$, $\langle F_w \rangle$ and $\langle F_A \rangle$ over the relative true wedge heights $\tilde{h}/h_{\max}$. The gray region in each figure indicates relative heights commonly found in real thermograms, adapted from Table 1.

In Figure 12a, $\langle F_N \rangle$ is plotted over the relative wedge heights $\tilde{h}/h_{\max}$. For a relative height $\tilde{h}/h_{\max} < 0.2$, no wedge is detected by the algorithm because it is too small. Therefore, no information about the positions or the areas is available and the thermograms are excluded from the analysis for the determination of the position error and the relative

deviation of the size and area of the wedge. For large wedges near $\tilde{h}/h_{\max} = 1$, $\langle F_N \rangle$ increases to 11%, which means that more than the correct number of wedges are found. This can be explained by the fact that the wedge templates are drawn from an interval of $[0.5, 0.95] \cdot h_{\max}$. When the templates are consistently smaller than the turbulence wedges, two maxima form in the cross-correlation curve close to the actual position of the wedge: One at the left edge of the wedge, one at the right, which leads to the algorithm detecting two smaller wedges close together instead of one large wedge.

Figure 12b shows the mean position error $\langle \frac{\Delta x}{\tilde{w}} \rangle$ over the relative height. Note that the mean wedge width $\langle \tilde{w} \rangle$ was calculated for each height parameter instead of averaging over all. For small wedges with $\tilde{h}/h_{\max} < 0.4$, the largest position error of $\langle \frac{\Delta x}{\tilde{w}} \rangle = 10\%$ can be found. However, for wedges with typical heights found in real thermograms (see gray region in Figure 12b), the position error is only around 1%. For large wedges near $\tilde{h}/h_{\max} = 1$, where more than one wedge is detected, the average position error only includes the position of the first found wedge. The position error therefore drops to $\langle \frac{\Delta x}{\tilde{w}} \rangle = -5\%$ for $\tilde{h}/h_{\max} = 1$, indicating that the first wedge is found slightly left of the actual position. This issue could be solved by using larger templates; however, this wedge size seldom occurs in real thermograms.

In Figure 12c, the relative deviation of the height, the width and the area of the wedge is shown over the true relative wedge height. For wedges with $0.2 < \tilde{h}/h_{\max} < 0.3$, the algorithm determines larger-than-true areas, which is evident from $\langle F_A \rangle > 0$. For wedges larger than $\tilde{h}/h_{\max} > 0.6$, the areas found are consistently smaller than the true areas, which is also apparent in the verification results in Table 2. This can be explained by the intensity gradient in the thermogram, which reaches the intensity value of the wedge at $\tilde{h}/h_{\max} = 0.73$. Therefore, the tips of the wedges in this height range have a CNR close to zero, which interferes with the determination of the wedges' areas.

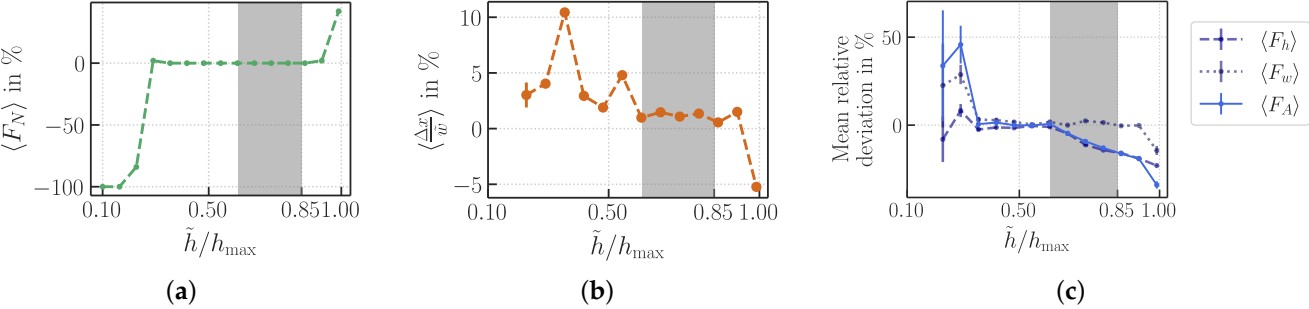

**(a)**          **(b)**          **(c)**

**Figure 12.** Analysis of the deviation of the height $h$, the width $w$ and the area $A$ of one wedge in simulated thermogram, averaged over $n = 50$ trials for each relative height value $\tilde{h}/h_{\max}$. The CNR stays constant at a value of 11. (**a**) shows the mean relative deviation of the number of detected wedges $\langle F_N \rangle$ over the relative true wedge heights $\tilde{h}/h_{\max}$. The mean normalized position error $\langle \frac{\Delta x}{\tilde{w}} \rangle$ over the true relative heights $\tilde{h}/h_{\max}$ of the wedges is shown in (**b**). The relative deviations of the height, the width and the area are shown in (**c**). The grey region indicates relative heights commonly found in real thermograms, i.e., $0.73 \pm 1\,\sigma$, with $1\,\sigma = 0.12$. The error bar indicates the standard deviation of the mean.

### 4.3. Validation: Application on Thermograms of an In-Service Wind Turbine

To validate the MBA, it was applied to real thermograms of rotor blades of in-service wind turbines. In Figure 13a, such a real thermogram of a wind turbine blade with five well-separated turbulence wedges with a CNR value of 6 or more is shown. Since the ground truth is not known, the measured areas can only be compared with a manually created reference, which is depicted in Figure 13b. The angle of the turbulence wedges is specified as 5°. The state-of-the-art results with the GBA are shown in Figure 13c for comparison; the results of the novel MBA are shown in Figure 13d. The GBA only detects the turbulence wedges with CNR > 15, resulting in only two detected turbulence wedges, while the MBA detects all five turbulence wedges, regardless of the CNR value. In Table 3, the values of the determined area, the reference area and the relative deviation of the total

area $F_A$ are displayed. Note that the deviation of the total area for the GBA results in $F_A = -64.2\%$, while the MBA only has a deviation of $F_A = -7.7\%$. The deviation value is in agreement with the deviation determined in the characterization of the algorithm in Section 4.2, where for CNR $\geq 15$, a deviation of $F_A \leq 8\%$ is expected, see Figure 11c.

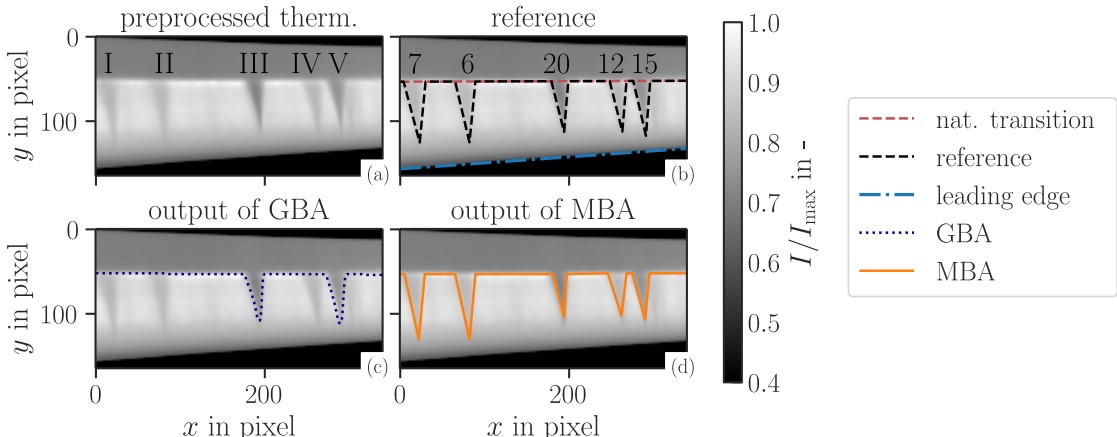

**Figure 13.** Thermogram of a segment of a rotor blade of an in-service wind turbine with $N = 5$ turbulence wedges. (**a**) shows the preprocessed thermogram with the turbulence wedges numbered with roman numerals. (**b**) shows the reference transition line, which was estimated manually, as well as the CNR of the wedges noted above the respective wedge. (**c**) shows the result of applying the gradient-based algorithm (GBA) of Gleichauf et al. [14] to the image and (**d**) shows the result of the application of the model-based algorithm (MBA) on the thermogram. Note the reduced color scale.

**Table 3.** Areas determined by the model-based algorithm (MBA) as well as the state-of-the-art, gradient-based algorithm (GBA) for all turbulence wedges in Figure 13. The area values are given in unit pixel$^2$ and are summed over all turbulence wedges. $N$ is the found or true number of wedges in the image.

|  | Total Wedge Area $A$ | $F_A$ in % | $N$ |
|---|---|---|---|
| Reference Value | 3814.5 | 0 | 5 |
| GBA | 1364.9 | $-64.2$ | 2 |
| MBA | 3520.5 | $-7.7$ | 5 |

In contrast to Figure 13, Figure 14a shows a wind turbine blade with nine smaller turbulence wedges that are close together or even touching. The skew of the turbulence wedges is also larger, so that the angle of the turbulence wedges amounts to $12°$. Furthermore, all turbulence wedges have a CNR value below 7. Between the seventh and eighth wedges, a small region with lower intensity might be visible that has a CNR of only CNR $= 0.2$, and was therefore excluded from the manually drawn reference line, seen in Figure 14b. The GBA again only detects turbulence wedges with CNR $> 3$, while the MBA again detects all turbulence wedges, see Figure 14c,d. This means that the GBA can detect wedges with a lower CNR in this thermogram than in the thermogram depicted in Figure 13. This is probably due to the preprocessing of the GBA, which applies a histogram equalization to each investigated thermogram. Therefore, different thermograms can be optimized to different degrees, leading to differences between thermograms with regard to the CNR of still detectable turbulence wedges. The MBA does not exhibit this fluctuation in the performance as no preprocessing is needed, which makes the MBA more reliable.

In Table 4, the summed areas of all wedges as determined by each algorithm and the manually estimated reference area are listed. Even though the GBA does not detect every turbulence wedge, the found areas of each wedge are determined to be much larger than they actually are due to the image noise, which leads to a remaining relative deviation of the total area of $-5.1\%$. However, a quantitative comparison of the determined area of each

wedge to the true area would show the error in the area determination more clearly. On the other hand, the MBA detects ten wedges, which includes, in addition to the nine referenced wedges, the turbulence wedge with CNR = 0.2 between wedges VII and VIII. However, the determined areas are always slightly smaller than the true areas of the wedges, which in total leads to a relative deviation of −20.2%.

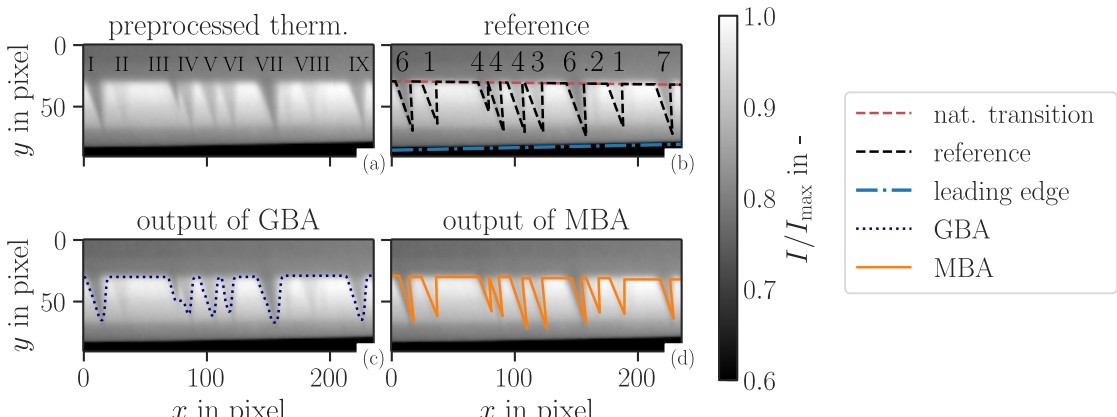

**Figure 14.** Thermogram of rotor blade of an in-service wind turbine with $N = 9$ turbulence wedges. (**a**) shows the preprocessed thermogram with the turbulence wedges numbered with roman numerals. (**b**) shows the reference transition line, which was estimated manually, as well as the CNR of the wedges noted above the respective wedge. (**c**) shows the result of applying the gradient-based algorithm (GBA) of Gleichauf et al. [14] to the image and (**d**) shows the result of the application of the model-based algorithm (MBA) on the thermogram. Note the reduced color scale, which makes the turbulence wedges more visible.

**Table 4.** Areas determined by the model-based algorithm (MBA) as well as the state-of-the-art, gradient-based algorithm (GBA) for all turbulence wedges in Figure 13. The area values are given in unit pixel$^2$.

| Area Type | Total Wedge Area $A$ | $F_A$ in % | $N$ |
|---|---|---|---|
| Reference Value | 2080.0 | 0 | 9 |
| GBA | 1972.9 | −5.1 | 6 |
| MBA | 1659.0 | −20.2 | 10 |

The validation demonstrates that the proposed model-based wedge detection algorithm is capable of detecting turbulence wedges and their areas on real thermograms of wind turbines in operation. The algorithm was able to determine wedges with CNR values in a range of $0.2 \leq \text{CNR} \leq 20$. The state-of-the-art algorithm consistently detects fewer wedges than actually present in the thermogram due to low CNR values, but overestimated the areas for each found wedge. While the model-based algorithm detected the number of wedges correctly, it underestimated the area by 7% to 20% depending on the CNR of the turbulence wedges, which is in agreement with results from the characterization of the algorithm.

## 5. Conclusions and Outlook

A model-based image processing algorithm for the detection of turbulence wedges on rotor blades of wind turbines in operation was introduced. The algorithm utilizes an image-processing approach called template matching, a technique for finding parts of a thermogram which match a template image. The templates are wedge-shaped, which imitates the natural shape of a turbulence wedge. Using the template-matching method, turbulence wedges are detected as a whole, including position and size parameters, instead of identifying single positions of the premature transition line, which is the case in the state-of-the-art gradient-based method. Therefore, the approach simplifies the determination

of the additional turbulent area on the rotor blade due to turbulence wedges, once the turbulence wedges are identified and located. The positions of the turbulence wedges between the natural transition line and the leading edge are detected first using a cross-correlation of the wedge templates and the thermogram. Then, the areas are determined by comparing the detected wedges to wedge templates of different sizes, where the size of the template with the highest similarity measure is finally considered as the measured size of the turbulence wedge.

The model-based algorithm was verified and compared to the state-of-the-art algorithm by evaluating a simulated thermogram with three turbulence wedges of different CNR values. The state-of-the-art algorithm was only able to detect one of the three wedges, leading to a relative deviation of the area of $-38.5\%$. The area of the single detected wedge was overestimated, which is why the relative deviation is not smaller. On the other hand, the model-based algorithm found all three and also estimated their areas with a relative deviation of 1.5%.

A detailed characterization of the algorithm was performed at different CNR values and different wedge sizes. For $4 \leq \mathrm{CNR} \leq 16$, which are CNR values commonly found in real thermograms, the absolute deviation of the determined position stayed below 2.5% of the average wedge width. The relative deviation of the wedge areas remains below 10% for $4 \leq \mathrm{CNR} \leq 16$. Overall, wedges with a CNR larger than 2 were detectable using the model-based algorithm.

To finally validate the model-based algorithm, it was applied to two real thermograms of rotor blades of wind turbines in operation. A comparison with the state-of-the-art algorithm shows an improvement in the number of wedges detected: the state-of-the-art algorithm consistently found fewer wedges than truly present in the thermogram, leading to relative deviations of the area of up to $-64.2\%$. However, the area of the detected wedges was often overestimated. The model-based algorithm found all wedges but in general underestimated their areas. Even so, the model-based algorithm had relative deviations of at most $-20.2\%$, which, in addition, can be reduced by correcting the systematic error that was determined according to the algorithm characterization. As a result, a clear improvement in the detection of turbulence wedges and their areas compared to the state-of-the-art algorithm was achieved. Hence, a more accurate analysis of the impact turbulence wedges have on the efficiency and the annual energy production of the wind turbine is enabled.

As an outlook, the detection of the turbulence wedges' positions still can be improved. This concerns, in particular, turbulence wedges near the edge of the thermogram, which currently have a lower probability of being detected, but which could be solved by padding the thermogram. Additionally, the determination of the area can be improved. Instead of using linearly decreasing weights for the weighted correlation evaluation, the actual intensity gradient in the thermogram (in particular at the wedge tip) needs to be automatically recognized and then taken into account in the weighting. Furthermore, the wedge detection algorithm can be enhanced to work with subpixel accuracy by using interpolated thermograms and templates. Lastly, an automatic determination of the wedge angle is desirable in the future so that the algorithm then works completely automatically. A further next step is the application of the algorithm for intensive field measurements of wind turbines in motion to investigate the additional area with turbulent flow due to the turbulence wedges over a one year operation and longer. On a shorter time scale, the algorithm will be used to study dynamic flow effects and how they influence the turbulence wedges.

**Author Contributions:** Conceptualization, A.-M.P., M.S. and A.F.; methodology, A.-M.P., D.G. and A.F. ; software, A.-M.P.; formal analysis, A.-M.P.; investigation, D.G.; writing—original draft preparation, A.-M.P.; writing—review and editing, A.-M.P., D.G., M.S. and A.F.; visualization, A.-M.P.; supervision, A.F. and M.S.; project administration, M.S.; funding acquisition, D.G., M.S. and A.F. All authors have read and agreed to the published version of the manuscript.

**Funding:** This research was funded by the German Federal Ministry for Economic Affairs and Energy (BMWi) within the project of PreciWind, grant number 03EE3013A.

**Conflicts of Interest:** The authors declare no conflict of interest.

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
