# Peer review of "Automated Detection of Premature Flow Transitions on Wind Turbine Blades Using Model-Based Algorithms"

_applsci, doi:10.3390/app11188700_

Round 1

Reviewer 1 Report

Minor addition required

  1. The paper talks about finding wedge templates. Need to elaborate if these templates are generated at identical atmospheric conditions (standard temperature and pressure).

Reviewer 2 Report

Line 124. Remove "Therefore".

Line 207-209. Can the authors justify and/or give a literature refernce for the selection of 0.1 as threshold value?

Line 279. Again, can the authors describe from where the empirical value of 35% comes from? If it has been derived from previous experience can they show how it was calculated?

Figure 4b. It is possible to insert a legend correlating the line color to the exact value of the template size?

Line 360-361. Please insert  a reference to investigation on h/w ratio and the relative applicability to the case explored in this study 
